# Prevalence and molecular characterization of antibiotic resistance and associated genes in *Klebsiella pneumoniae* isolates: A clinical observational study in different hospitals in Chattogram, Bangladesh

**Afroza Akter Tanni**[1], **Md. Mahbub Hasan**[1], **Nahid Sultana**[2], **Wazir Ahmed**[3], **Adnan Mannan**[1]*

**1** Department of Genetic Engineering & Biotechnology, Faculty of Biological Sciences, University of Chittagong, Chattogram, Bangladesh, **2** Department of Microbiology, Chattogram Maa O Shishu Hospital, Agrabad, Chattogram, Bangladesh, **3** Department of Neonatology, Chattogram Maa O Shishu Hospital, Agrabad, Chattogram, Bangladesh

* adnan.mannan@cu.ac.bd

## Abstract

### Objective

This study was performed to investigate the prevalence of multidrug resistance and molecular characterization of *Klebsiella pneumoniae* (KPN) from clinical isolates in the southern region of Bangladesh. Additional analysis of the prevalence of $bla_{NDM-1}$, $bla_{SHV-11}$, $uge$ genes of KPN was also carried out among these clinical isolates.

### Method

The study was carried out using 1000 clinical isolates collected from two different hospitals of Chattogram. A drug susceptibility test was performed by the disk diffusion method to detect KPN's response to 16 antibiotics. The presence of antibiotic-resistant and (or) virulent genes $bla_{NDM-1}$, $bla_{SHV-11}$, $uge$ were investigated using the PCR technique. Isolates having $bla_{NDM-1}$, $bla_{SHV-11}$, $uge$ gene were further validated by sequencing followed by phylogenetic analysis. Phylogenetic relationships among these isolates were determined by Clustal omega and MEGA7.

### Result

A total of 79%, 77%, 74.9%, 71%, 66% and 65% isolates exhibited resistance against cefuroxime, cefixime, cefotaxime, ceftazidime, cefepime and ceftriaxone respectively. The frequency of resistance to other antibiotics varied from 26.5% to 61.8%. PCR analysis showed that 64% of strains harbored $bla_{NDM-1}$ gene, and 38% strains harbored $bla_{SHV-11}$ gene. Moreover, 47% of samples were carrying $uge$ gene, and 19% of samples carried $bla_{NDM-1}$, $bla_{SHV-11}$, $uge$ genes together.

**Data Availability Statement:** All relevant data are within the manuscript and its Supporting Information files.

**Funding:** This study was partially funded by Research and publication office, University of Chittagong (Award Number: 6427/Re/Plan/Pub/Div/CU/2018). The funders had no role in study design, data collection and analysis, decision to publish, or preparation of the manuscript. There was no additional external funding received for this study.

**Competing interests:** The authors have declared that no competing interests exist.

## Conclusion

In this study, we've analysed the pattern of expression as well as prevalence of bla$_{NDM-1}$, bla$_{SHV-11}$, and uge genes in Klebsiella isolates. Upon molecular and statistical analysis, we found a high prevalence of multi-drug resistance KPN strains in the isolates. The *Klebsiella* isolates were confirmed to harbor multiple ESBL genes and 64% of the isolates were found to be producing NDM-1. As multidrug resistance is an alarming issue, continuous surveillance and routine clinical detection of resistant bacteria and plasmids are necessary to prevent catastrophic public health incidents.

## 1 Introduction

Since the discovery and commercialization of antibiotic drugs, preventing and treating pathogen mediated and driven diseases have become comparatively more accessible. However, this ceaseless use of antibiotic drugs has been subject to abuse as well through irregular administration and overdose in veterinary, livestock breeding, and farming [1–3], which has caused multiple pathogens to develop resistivity against antibiotics. Over the years antimicrobial resistance (AMR) has become one of the major concerns of global health issues [4].

Given the gravity of the AMR issue, data relevant to AMR is being collected and analyzed constantly through clinical studies in multiple populations. Yet, according to a study by Ahmed *et al*. (2019), major gaps were identified in South Asia, African and Eastern Mediterranean regions on global surveillance of AMR, data collection, sharing and data coordination, method standardization [5]. In recent years, the number of resistant strains has increased, due to causes including an increase in antibiotic usage, inter-generic and inter-specific conjugal transmission of antibiotic-resistant genes between bacteria, and selective pressure [6]. While patterns in AMR can differ within bacterial genera and species, in Bangladesh, *Enterobacteriaceae* were found to be resistant to the following classes of antibiotics: aminoglycoside, macrolide, and beta-lactams; which was confirmed through analyzing clinical isolates from patients admitted to various tertiary hospitals with acute respiratory infections, wound infections, typhoid fever or diarrhea. In regards to the nature of the genetic aspects of AMR in Bangladesh, the matter is still under observation and the data is limited [7].

*Klebsiella pneumoniae* (KPN), a member of multidrug-resistant ESKAPE pathogens groups (*Enterococcus faecium*, *Staphylococcus* aureus, *Klebsiella pneumoniae*, *Acinetobacter baumannii*, *Pseudomonas aeruginosa*, and *Enterobacter* species) [8], is an agent of both nosocomial and community-acquired infections. Its high association has been found with urinary tract infections, pneumonia, septicemia, burns, pyogenic liver abscesses, wound infections, meningitis, endophthalmitis, and lung abscess [9–13]. KPN strains acquire a multidrug resistant phenotype through horizontal transfer of antimicrobial resistance genes carried by either transposons or plasmids. The transfer is usually mediated by mobile genetic elements, Integrons (IncFII, IncN and IncI1) [14–16]. In Bangladesh, a dramatic increase of β- lactam resistant KPN was observed from 2001 to 2011 [17–19]. The situation became more severe with the discovery and spread of the novel carbapenemase, New Delhi Metallo-ß-lactamase (NDM), proteins that give bacteria the ability to resist multiple types of antibiotics, rendering superbugs. Among 5 variants of NDM (NDM-1 to NDM-5), NDM-1 is endemic in India and also common in South Asian countries such as Pakistan, Nepal and Bangladesh [20–22]. In Bangladesh, NDM-1 has been found significantly prevalent in natural water samples, sewers and even in clinical samples [7, 23, 24].

Because of having a single copy of chromosomal gene $bla_{SHV-11}$ that code for SHV-11 and its derivatives which is a class A beta-lactamase, KPN actively and predominantly resists multiple classes of β-lactams [25]. The mutation in the promoter region of the gene, $bla_{SHV-11}$ changes the enzyme's affinity and increases enzyme production or diminishes Penicillin-binding proteins (PBP) located in the cytoplasmic membrane [26–30]. SHV -11 was identified in KPN derived from clinical isolates in Bangladesh [31] and some novel alleles of SHVs were obtained in clinical isolates in Mymensingh. KPN isolates, carrying SHV type ESBL genes were found in natural water samples from lakes in Dhaka [24]. As for the patient samples, isolates from urine and tracheal aspirates contained a prominent distribution of KPN virulence factors. Among those, a capsule associated gene, UDP galacturonate 4-epimerase (*uge*) is commonly found in KPN and promotes infection by resisting phagocytosis [12]. Surface expression of smooth lipopolysaccharide (LPS) and capsular polysaccharide with K antigen is typically found in wild type KPN isolates. Noteworthy to mention, mutations in the *uge* gene creates a mutant strain with O-:K- phenotype, lacking a capsule and LPS without O antigen molecules devoid of the outer core oligosaccharide. The mutant strain also transforms UDP-glucose (UDP-GlcA) to UDP-galacturonic acid (UDP-GalA) by uridine diphosphate galacturonate 4-epimerases (UDPGLEs) enzymatic activity, rendering a rather avirulent KPN [32, 33].

This study aimed to investigate the antibiotic resistance pattern of KPN in the southern part of Bangladesh through analyzing clinical samples obtained from neonates and adults. We also investigated the prevalence of extended-spectrum beta-lactamases (ESBLs) genes of clinical isolates of MDR KPN in southern Bangladesh.

# 2 Materials and methods

## 2.1 Study setting

A cross-sectional study was conducted in two healthcare centers in Chattogram to collect clinical samples including blood, urine, pus, tracheal aspirates, throat swab, umbilical swab, sputum, and wound swab, high vaginal swab (HVS) with traces of KPN from both outdoor and indoor patients. The tenure of sample collection was from August 2018 to November 2019. Samples were collected from various hospital wards such as Medicine, Gynae, Neonatal and Pediatric surgery, Adult surgery, Special care baby unit, Neurology, Diarrhoea, Thalassemia, Orthopedics, Intensive Care Unit (ICU), HDU (Child), Neonatal Intensive Care Unit (NICU). The demographic and epidemiological data of patients with KPN infections were collected from medical records and microbiology databases owned and dispatched by the microbiology laboratory of Chattogram Maa-Shishu O General Hospital Microbiology lab, and CHEVRON hospital and Clinical Laboratory (PTE) Ltd. After screening, we have finally included 1000 samples with confirmed *K. pneumoniae* (KPN) infections.

## 2.2 Ethical approval

The study protocol was ethically approved by the institutional review board of Chattogram Maa-Shishu O General Hospital Medical College (Ref: CMOSHMC/IRB/2018/5). Oral consent was obtained from the patients or from parents or guardians (in case of minors). Need for written consent was waived by the ethics committee.

## 2.3 Confirmation of *K. pneumoniae* (KPN) isolates

Presence of KPN was confirmed through analyzing microbial cultures, Gram staining and conventional biochemical tests according to CLSI (Clinical Laboratory Standard Institute) guidelines. Initially, the bacterial colony and shape was checked in Mac-Conkey agar medium,

Blood agar and Chocolate agar. Gram staining was done for bacterial group differentiation. Once the presence of KPN in clinical samples were confirmed, a number of conventional biochemical tests were performed [Indole, Methyl red, Voges Proskauer, Citrate, TSI (triple sugar Iron, Motility and Urease tests)] using subcultures for of *Klebsiella pneumoniae* isolates [34–36].

## 2.4 Determination of antibiotic susceptibility pattern of KPN

The Kirby-Bauer method (KB) was adapted to conduct antibiotic susceptibility tests for KPN obtained from clinical samples. An overnight culture of KPN adjusted to standard suspension of isolate confirming 0.5 McFarland turbidity was inoculated on the surface of two Mueller Hinton agar (HIMEDIA, India) plates using selective antibiotic disks, namely Amikacin (30 μg), Amoxi-clav (30 μg), Ampicillin (30μg), Cefepime (30μg), Cefixime (5μg), Ceftazidime (30μg), Ceftriaxone (25μg), Chloramphenicol (30μg), Ciprofloxacin, (5μg), Cotrimoxazole (23.75μg), Gentamicin (10μg), Imipenem (10μg), Levofloxacin (5μg), Meropenem (24μg), Netilmicin (10μg), and Piperacillin (30μg) according to CLSI guidelines [36]. For negative control, a blank disk of filter paper was used. Multidrug resistant samples were selected for molecular analysis. Isolates showing resistance to highest number of antibiotic categories were chosen for further analysis.

## 2.5 Molecular detection of $bla_{NDM-1}$, $bla_{SHV-11}$ and *uge*

The $bla_{NDM-1}$, $bla_{SHV-11}$ and *uge* genes from 100 confirmed MDR KPN isolates were identified through polymerase chain reactions (PCR) using the following sets of primers:

1. bla-$_{NDM-1}$: 5′–GGTTTGGCGATCTGGTTTTC–3′; 5′–CGGAATGGCTCATCACGATC–3′ (annealing temperature 51.2˚C) [12, 21].

2. bla$_{SHV-11}$: 5′–ATGCGTTATATTCGCCTGTGTATT–3′; 5′–GCGTTGCCAGTGCTCGATC AGCGC–3′ (annealing temperature 51.2˚C) [37, 38].

3. uge: 5′–ATGCGTTATATTCGCCTGTGTATT–3′, 5′–GCGTTGCCAGTGCTCGATCAGCG C–3′ (annealing temperature 52.3˚C) [12].

Genomic DNA of KPN was extracted using the Boiling method [39, 40]. Sanger sequencing was carried out with the purified PCR products of a total of 15 isolates (5 isolates from each group carrying $bla_{NDM-1}$, *blaSHV-11*, *uge* gene). All draft sequences were aligned using BioEdit Sequence Alignment Software (version 7.0.5.3) followed by nucleotide BLAST for further confirmation and to identify nucleotide variations. After analysis, all sequences were deposited to the NCBI database. The accession numbers of SHV-11 gene containing KPN isolates are MN437452, MN551175, MN551176, MN551177 and other sequences have been submitted to NCBI. Before phylogenetic analysis, all obtained nucleotide sequences were translated through the EMBOSS Transeq program. Following translation, the protein sequences were aligned using Clustal Omega (https://www.ebi.ac.uk/Tools/msa/clustalo/).

## 2.6 Phylogenetic tree construction

After alignment, a phylogenetic tree was constructed using MEGA7 using the maximum likelihood method [41]. The evolutionary history was inferred by using the Maximum Likelihood method based on the JTT matrix-based model. Initial tree(s) for the heuristic search were obtained automatically by applying Neighbor-Join and BioNJ algorithms to a matrix of pairwise distances estimated using a JTT model, and then selecting the topology with a superior log-likelihood value. The tree is drawn to scale, with branch lengths measured in the number of substitutions per site.

## 3 Results

### 3.1 Characteristics of KPN isolates

We analyzed a total of 1000 KPN isolates from two healthcare facilities in this study. The male to female ratio of the participants was found to be1:1.13. By analyzing age groups, children (<15 years age) represented 47.6% of the total study population whereas, young adolescents (15–29 years), adults (30–44 years) and elder age group (≥45 years) covered 18.6%, 10.9% and 22.9%, respectively (Table 1). About 46.5% of the samples were collected from outpatients and 53.5% were from inpatients with KPN infections of which 21.87% were from medicine wards, 19.25% were from the Gynaecology ward, 9.72% were from pediatric surgery wards. Confirmed KPN isolates were collected from a wide variety of samples including urine (39.9%), tracheal aspirates (14.2%), pus (14.1%), blood (13.8%), sputum (10.1%) and others (7.9%) (Table 1). Multidrug resistance was more among the male (74.6%). The prevalence of MDR observed in the urine, tracheal aspiration, blood, pus, sputum and wound swab samples was recorded: 65.1%, 81.25%, 78.2%, 65.9%, 51.4% and 65.9% respectively.

### 3.2 Antimicrobial resistance patterns of KPN

The majority of KPN isolates were confirmed as multidrug-resistant in the disk diffusion susceptibility test. Most of the isolates were resistant against cefuroxime (79%), cefixime (77%), cefotaxime (74.6%), ceftazidime (71%), cefepime (66%) and ceftriaxone (65%) (Fig 1A) whereas, most effective antibiotics against KPN were amikacin, meropenem and imipenem with the resistance of 26%, 27% and 30%, respectively. Isolates from females showed more resistance to the antibiotics available in the market compared to those from males (Fig 1B).

**Table 1. Clinical characteristics of study population.**

| Variable | Total Cases (%) (n = 1000) | Sensitive (%) (n) | MDR (%) (n) | SDR (%) (n) | P value |
|---|---|---|---|---|---|
| *Gender* | | | | | |
| Male | 43% (430) | 6.97% (30) | 74.65% (321) | 18.37% (79) | 0.0029 |
| Female | 57% (570) | 10.87% (62) | 63.68% (363) | 25.43% (145) | |
| *Age (in years)* | | | | | |
| <15 years | 47.6% (476) | 3.99% (19) | 69.12% (329) | 26.89% (128) | 0.2151 |
| 15- <30 years | 18.6% (186) | 11.83% (22) | 70.43% (131) | 17.74% (33) | |
| 30- <45 years | 10.9% (109) | 19.27% (21) | 62.38% (68) | 18.35% (20) | |
| 45- <60 years | 13.0% (130) | 12.31% (16) | 69.23% (90) | 18.46% (24) | |
| ≥ 60 years | 9.9% (99) | 14.14% (14) | 66.67% (66) | 19.19% (19) | |
| *Patient type* | | | | | |
| Outdoor | 46.5% (465) | 31.18% (145) | 64.73% (301) | 25.59% (119) | 0.0215 |
| Indoor | 53.5% (535) | 8.79% (47) | 71.58% (383) | 19.63% (105) | |
| *Sample types* | | | | | |
| Urine | 39.9% (399) | 12.03% (48) | 65.16% (260) | 22.81% (91) | <0.001 |
| Tracheal aspirates | 14.2% (142) | 1.41% (2) | 85.21% (121) | 13.38% (19) | |
| Pus | 14.1% (141) | 9.93% (14) | 65.96% (93) | 24.11% (34) | |
| Blood | 13.8% (138) | 7.25% (10) | 78.26% (108) | 14.49% (20) | |
| Sputum | 10.1% (101) | 11.88% (12) | 51.48% (52) | 36.63% (37) | |
| Wound swab | 4.7% (47) | 6.38% (3) | 65.95% (31) | 27.66% (13) | |
| Others | 3.2% (32) | 12.5% (4) | 56.25% (18) | 31.25% (10) | |

MDR = Multidrug-resistant, SDR = Single or two drug-resistant

*** P value has been compared between multi drug resistant and single drug resistant samples.

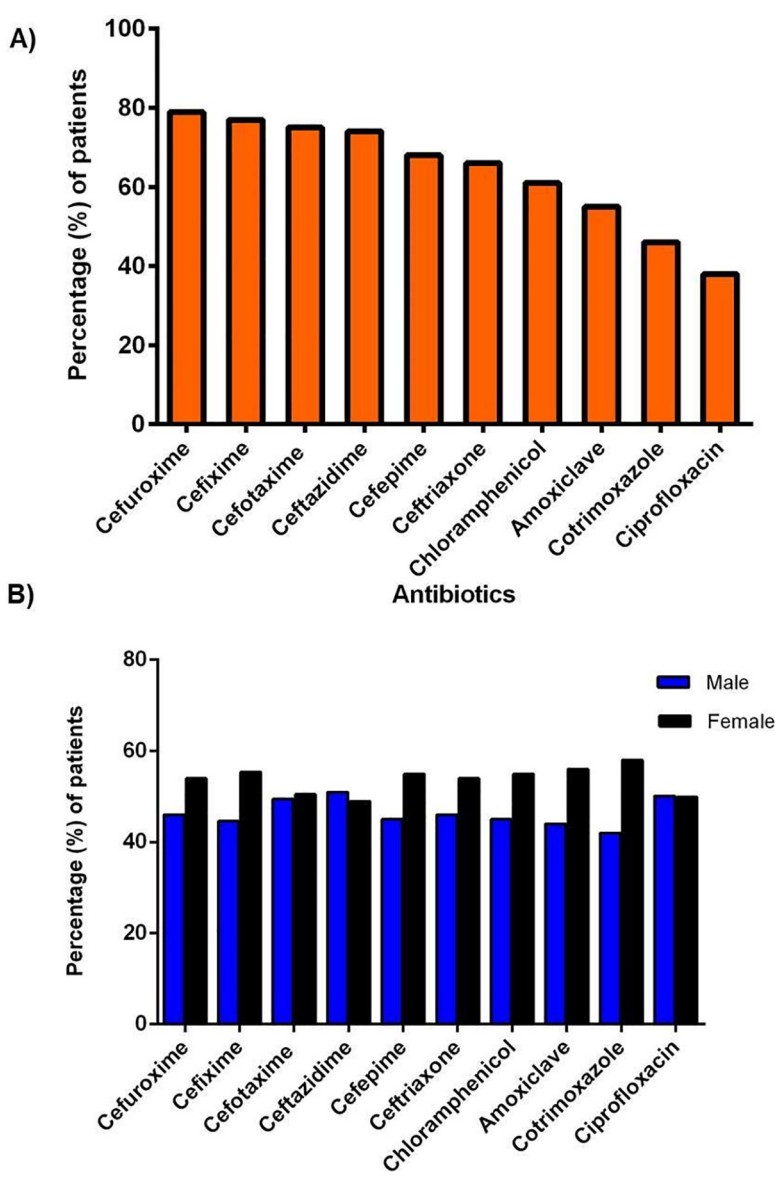

**Fig 1. Patterns of antimicrobial resistance among *Klebsiella pneumonia* isolates collected from two hospitals.** (A) Most resistant 10 antibiotics among study samples; (B) Gender wise frequency of antimicrobial resistance in KPN isolates.

Patients were most sensitive to nitrofurantoin (77.6%), amikacin (71.2%) and meropenem (68.6%) (Fig 2A), Isolates from both indoor and outdoor patients were most resistant to cefixime, ceftriaxone and cefepime (Fig 2B).

## 3.3 Prevalence of *bla*$_{NDM-1}$, *bla*$_{SHV-11}$ and *uge* among multi-drug resistant KPN isolates

Top 100 multidrug-resistant *KPN* isolates were selected and screened for determining the prevalence of *bla*$_{NDM-1}$, *bla*$_{SHV-11}$ and *uge* genes by PCR (**Fig 3**). 64% of selected KPN isolates were found *bla*$_{NDM-1}$ positive. About 48% of isolates were *uge* positive and 38% were *bla*$_{SHV-11}$

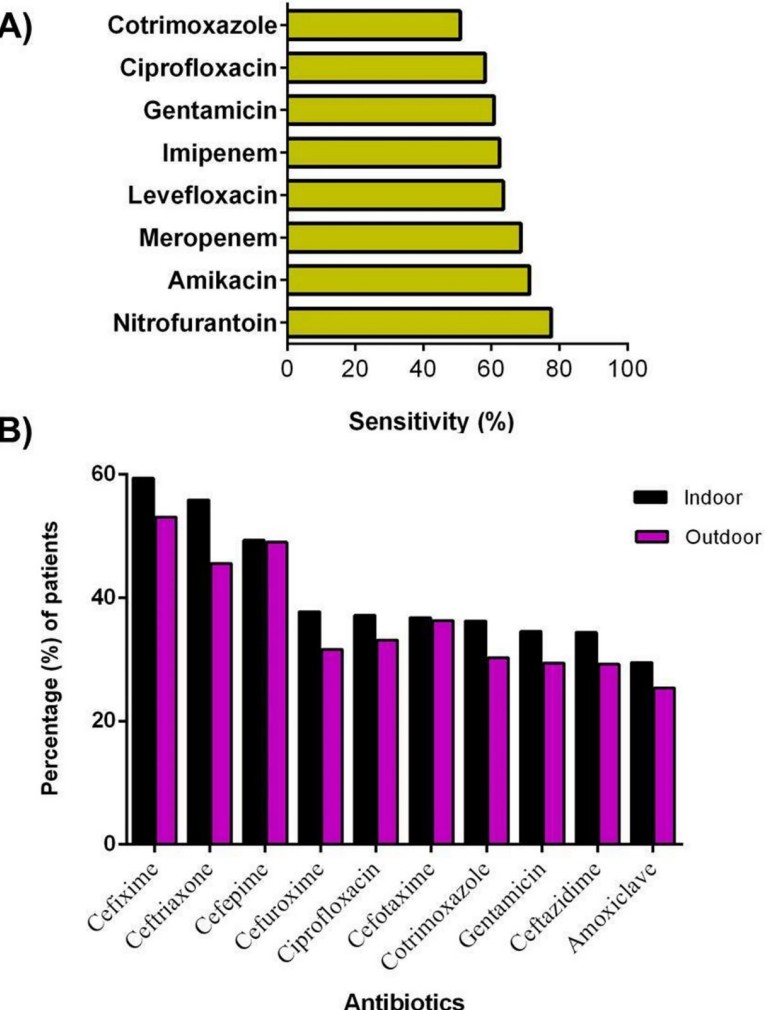

**Fig 2. Antibiotic sensitivity and resistance among different types of isolates.** A) Antibiotic sensitivity among patients (n = 1000); B) Antibiotic resistance according to origin (indoor and outdoor patients).

positive strains (**Table 2**). A total of 45% of KPN strains having $bla_{NDM-1}$ collected from urine samples, 24.2% from tracheal aspirates followed by blood (10.6%), pus (10.6%) and sputum (3%). Most of the patients infected with KPN infections encoding $bla_{NDM-1}$ were inpatients (65%), whereas 35% were outpatients (**Table 2**). The $bla_{SHV-11}$ gene was found in urine

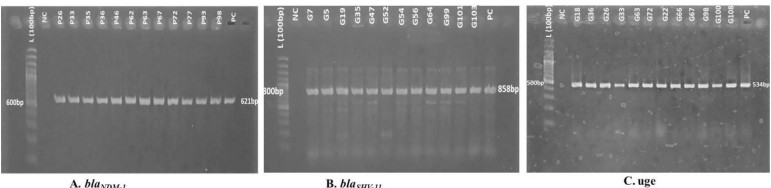

**Fig 3. Amplification and confirmation of $bla_{NDM-1}$, $bla_{SHV-11}$, and uge genes by polymerase chain reaction.** PCR products were visualized in 1.5% agarose gel using ethidium bromide staining (0.5%). L indicates Ladder (100 bp); NC as a negative control(Nuclease free water); PC indicates positive control, *K. pneumoniae* ATCC70603. *(A)* samples representing bands with 621bp were $bla_{NDM-1}$ positive amplicons similarly, *(B)* samples were $bla_{SHV-11}$ positive amplicons with 858bp, *(C)* samples were uge positive amplicons having 534bp.

Table 2. **Prevalence of $bla_{NDM-1}$, $bla_{SHV-11}$, $uge$ in top multi-drug resistant KPN samples (n = 100).**

| Genes | Percentage of positive Isolates | Gender | | Patient type | |
|---|---|---|---|---|---|
| | | **Male** | **Female** | **Outdoor** | **Wards/Indoor** |
| $bla_{NDM-1}$ | 64 | 30 | 34 | 27 | 37 |
| $uge$ | 47 | 24 | 23 | 17 | 30 |
| $bla_{SHV-11}$ | 38 | 21 | 17 | 20 | 18 |

samples at an increased rate (45%) as compared to tracheal aspiration (27.5%), blood (3.03%), pus (4.5%), throat swab (4.5%), sputum (2%) and wound swab (2%) (Fig 5). The $bla_{SHV-11}$ gene was found at an approximately equal ratio in KPN strains of both outpatients (52.7%) and inpatients (47.3%). Similarly, 47% of KPN isolates had $uge$ where 36.2% and 63.8% isolates were from outdoor and indoor patients respectively (Table 2). The $uge$ gene was found in urine (46.8%), tracheal aspirates (27.6%), blood (6.4%), pus (10.6%), sputum (6.4%) (Fig 5). 19 out of 100 isolates contained all three genes $bla_{NDM-1}$, $bla_{SHV-11}$ and $uge$. In clinical samples, $bla_{SHV-11}$ and $uge$ were more prevalent in males (51.17% and 55.3%, respectively) than in female patients. While $bla_{NDM-1}$ gene was prevalent in females (52.3%).

For further validation of gene frequency observed in PCR experiments, randomly, 5 PCR amplicons were subjected to gene sequencing. The raw sequences of $bla_{NDM-1}$, $bla_{SHV-11}$, $uge$ genes were then confirmed by comparing them with the reference sequence database using phylogenetic analysis (**Fig 4**).

### 3.4 Plasmid size determination

All tested isolates were screened for the presence of plasmid. 81 out of 100 isolates were found to have plasmids with a band size of 1500 bp (Fig 6). All plasmids were similar in size which was confirmed by agarose gel electrophoresis.

## 4 Discussion

Although KPN is a common inhabitant of intestinal microflora, it can metastasize outside the gut triggering a wide range of infections, predominantly in humans. Apart from its metastasizing capacity, what establishes KPN's reputation as a deadly pathogen is that over time, KPN has been evolving as a superbug. Previous studies have shown that KPN has acquired resistance to carbapenem and other β lactam antibiotics. The results of our study align with the previous studies, in which case, we also found that a high number of KPN isolates showed resistance to cefuroxime, cefixime, cefotaxime, ceftazidime, cefepime, and ceftriaxone. Among total isolates, the majority of the isolates were highly resistant to cefuroxime and the percentage of resistance was higher compared to other antibiotics (Fig 1). A similar finding was observed by Akhter *et al*. (2016) and Hossain *et al*. (2017), demonstrating a significant number of *Klebsiella* isolates resistant to ceftriaxone along with ampicillin and cotrimoxazole [42, 43]. These incidences of resistance were linked to unclean food, contaminated bed sheets, personal hygiene affecting breast-feeding and negligence in washing hands. In all cases, contaminated hands played a significant role in spreading the KPN isolate and promoting its evolution. However, in terms of resistance to broad-spectrum antibiotics, some studies found KPN to be resistant to several more antibiotics apart from the aforementioned [44]. Uddin *et al*. (2011) observed that *Klebsiella* spp. were highly resistant to tetracycline, rifampin, nalidixic acid, and 100% sensitive to imipenem. Yasmin *et al*. (2015) [45] also found that *Klebsiella* isolates were amoxicillin and nalidixic acid resistant [46]. In contrast, we found the isolates of our study were mostly imipenem, meropenem and amikacin sensitive. We observed an equal

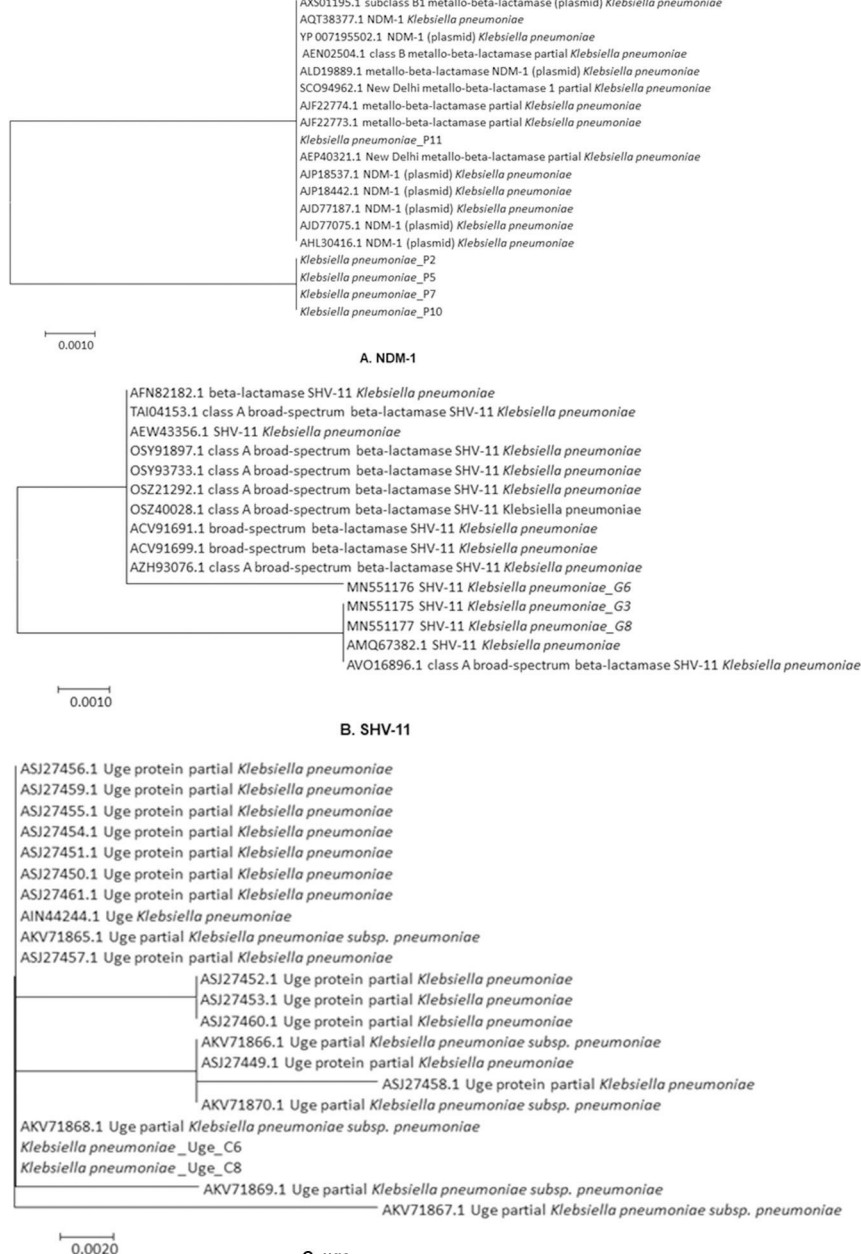

**Fig 4. Phylogenetic analyses of *NDM-1*, *SHV-11* and *Uge* genes from multidrug-resistant KPN isolates by maximum likelihood method.** The evolutionary history was inferred by using the Maximum Likelihood method based on the JTT matrix-based model. Three genes are A) NDM-1, B) SHV-11, C) uge.

distribution of infections in both males and females rin maximum age groups (Table 1). The 50 years male cohort was found to be more prone to the *Klebsiella* infections [47]. But the incidence of KPN infections was higher among the neonates and children that were included in the study sample. Frequent use of assisted ventilation and extensive administration of antibiotics in first-month post-birth in neonates, naive immune state, immunocompromisation, impairment of gut microflora could be some of the significant causes owing to their susceptibility to KPN infections which led to pulmonary infections and septicemia [42, 48].

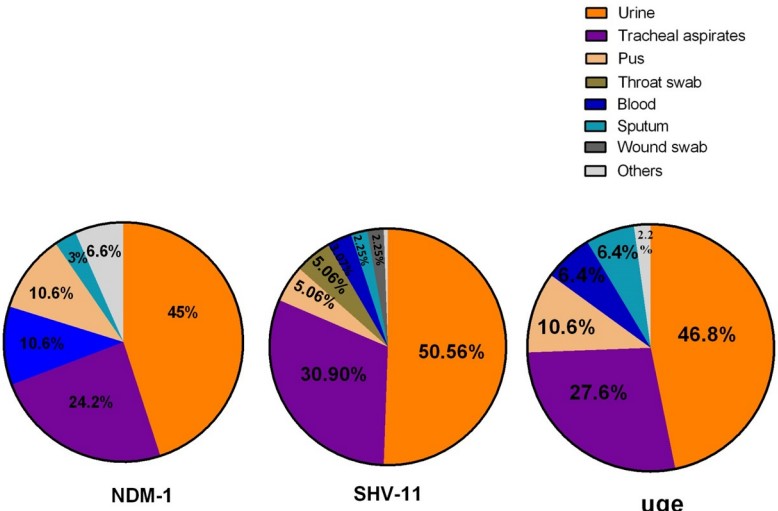

**Fig 5. Frequency of NDM-1, SHV-11 and uge genes in different types samples of *Klebsiella* isolates.**

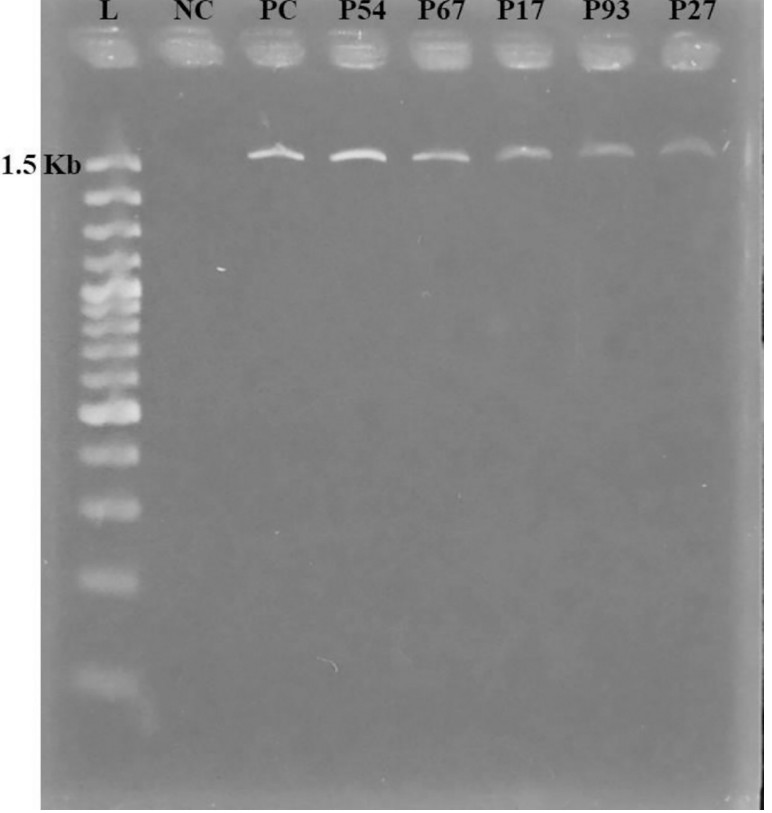

**Fig 6. Detection of plasmid by agarose gel electrophoresis.** Agarose gel electrophoresis (2%) was used for separation of the whole plasmid. Here, M indicates Marker (100 bp); N as the negative control, Nuclease free water; PC indicates positive control, *Klebsiella pneumoniae* containing the plasmid. All positive isolates had a band size of 1500 bp.

Choudhury *et al*. (2018) stated that 69.2% KPN isolates were the most common gut colonizer in neonates [49]. Considering the overall frequency, the incidence of KPN infections was found higher in patients admitted to the Medicine ward, Gynae ward, Pediatric surgery and neonatal ward among hospitalized patients in our study (Table 1). Urine was found to be the most common reservoir of KPN (39.9%) (Table 1). In a previous study, it was found that KPNs majorly producing ESBLs were isolated from urine samples [43]. Considering our results and that of Hossain *et al*. (2016), it can be considered that urine is indeed a predominant reservoir for KPN sustenance [43]. Presence of KPN in non-biological samples might be a probable explanation of the phenomenon of horizontal transfer of transposable elements into KPN isolates and rapid transmission of a plasmid to another *Enterobacteriaceae* [50, 51].

Carbapenem-resistant KPN is considered as a threat worldwide because of the rapid transmission of the plasmid-mediated $bla_{NDM-1}$ gene in *Klebsiella* species and the global spread of NDM-1 producing KPN. According to this study, the scenario is also similar in Chittagong City. In the present observation, 64% of KPN isolates were $bla_{NDM-1}$ positive (Fig 2A) (Table 2). NDM-1, belonging to the Class B Metallo-ß-lactamase (MBL) superfamily mediates the catalytic reaction. It has one or two catalytic zinc or iron binds to the active site and cleaves the amide bond of the beta-lactam ring, thus inactivating the antibiotics [52]. The majority of the $bla_{NDM-1}$ positive isolates were isolated from female patients than makes (64 vs 34) (Table 2). The incidence of KPN harboring $bla_{NDM-1}$ positive isolates was comparatively high in urine (30%) and tracheal aspiration (16%) (Fig 5). Bora and Ahmed (2012) found 8.67% $bla_{NDM-1}$ positive KPN isolates in clinical samples in northeast India [53]. 3.5% of NDM-1 producing *Enterobacteriaceae* was found in clinical samples in the Clinical Microbiology Laboratory of the International Centre for Diarrheal Disease Research, Bangladesh (ICDDR,B) [37]. In another study, 71% of isolates were $bla_{NDM-1}$ positive in hospital adjacent areas and 12.1% were positive in the community areas where KPN was prevalent in sewage samples [23]. Colonization of NDM-1 carrying KPN isolates in patients gut and long term hospitalization is also a risk for frequent *Klebsiella* infections. Plasmid-mediated antibiotic-resistant genes transferred to *Klebsiella* species or from other gram-negative bacteria or by the transposable elements may be the reasons for the rapid transmission of NDM-1 in environmental samples [20].

The chromosomal $bla_{SHV-11}$ was identified in hospitalized and outdoor patients at an equal ratio. Khan *et al*. (2018) identified 27% isolates producing SHV variants (SHV-201 and SHV-202) from clinical samples. SHV-12 was found in MDR KPN isolates from surface water [24]. The risk factors for the spread of ESBL positive isolates may be prolonged hospitalization, stay in the ICU, invasive entry of ESBL positive isolates through insertion endotracheal, urinary and central venous catheters. In the present study, 48% KPN isolates containing virulence gene, *uge* were derived and prevalent in urine (54%) and tracheal aspiration (20%). It was similar to the findings of Candan & Aksöz (2015) [12], and defined that there was no correlation among virulence factors and antibiotic-resistant genes, but they both contribute to bacterial pathogenesis. Common sources of KPN and cross-contamination among patients may be the main reasons for the rapid transmission of the *uge* gene. In our study, 34% KPN isolates were found to harbour NDM-1 and *uge* in the same strain derived from clinical samples and 19% KPN isolates were defined to carry all tested genes, $bla_{NDM-1}$, $bla_{SHV-11}$, and *uge*.

In this study, 81 plasmids of 1400 bps in size were isolated from various clinical samples (Fig 6). A similar result was characterized by Balm *et al*. (2013), where plasmids of different types and sizes were identified [54]. The size of the plasmids obtained from previous study were ~ 1.4kb in the KPN isolates from urine, sputum samples, and ranged from ~ 9kb to ~ 194kb in blood, tracheal aspirates samples. But another study stated that KPN identified in isolates obtained in the UK, India, and Pakistan had different $bla_{NDM-1}$ positive plasmids with

different sizes [42]. Islam *et al.* (2013) reported that harbouring plasmid size ranges from 60 to 100 MDa in KPN and also observed that KPN isolates, causing urinary tract infections to have a plasmid size of 9.8 Kb [20]. The source of plasmid dissemination or spread of KPN in a single hospital or same environment may be the fact of having the same plasmid in KPN isolates. The plasmid-mediated $bla_{NDM-1}$ gene and chromosomal $bla_{SHV-11}$ showed resistance to carbapenems and fifth-generation of cephalosporins due to the overproduction of NDM-1 enzymes, alteration of penicillin-binding proteins (PBPs), changing cellular permeability and increasing efflux pump [55].

This study found a high prevalence of antibiotic resistance gene $bla_{NDM-1}$ along with uge and $bla_{SHV-11}$ in KPN strains from isolates in the hospital. Therefore, to obstruct the spread of resistant strains, it is important to take measures for controlling *K. pneumoniae* infections. In the context of Bangladesh, establishing a national antibiotic resistance surveillance network, scrutinizing susceptibility to carbapenems, determination of clonal relationship based on geographical or epidemiological distribution by pulse-field gel electrophoresis analysis, plasmid conjugation analysis and working with larger samples will allow us to further analyse the patterns of antibiotic-resistant genes in isolates as well as the pool of antibiotics that KPN is resistant to. For further investigations, it is also necessary to analyse the protein products, and other subsidiary cis and trans activating mechanisms associated with $bla_{NDM-1}$, $bla_{SHV-11}$ and *uge*.

## 5 Conclusion

In the *K. pneumoniae* infection landscape, our study was the first to run an extensive molecular analysis on ESBL producing *K. pneumoniae* isolates from patients seeking medical care in different hospitals in Chattogram, Bangladesh. Through this study, we were able to identify early occurrences of NDM-1 producing organisms as well as prevalence of blaNDM-1 plasmid. The prevalence of blaNDM-1 plasmid indicates a large-scale distribution of NDM-1 producing bacteria in the community. A substantial increase in resistant KPN strains in the environment, as well as in a healthcare setting, increases the possibility of nosocomial infections; and through horizontal gene transfer, it could also amplify the persistence of the whole antibiotic-resistance situation. Rigorous screening for NDM-1, SHV-11, UGE and other MBLs in ESBL-producing organisms, because it provides insight into a more genotypic aspect on KPN antibiotic resistance along with significant epidemiological factors will help to formulate and implement plans for controlling the use and abuse of antibiotics on a large scale as well as substitute treatment options in resistant cases.

## Supporting information

**S1 Dataset.**
(XLSX)

**S2 Dataset.**
(XLSX)

**S1 Raw images.**
(PDF)

## Acknowledgments

The authors would like to thank the research assistants of Disease Biology and Molecular Epidemiology Research Group, Chattogram for their support during the study. We acknowledge support from the Research and Publication Cell, University of Chittagong in this research.

## Author Contributions

**Conceptualization:** Md. Mahbub Hasan, Wazir Ahmed, Adnan Mannan.

**Data curation:** Md. Mahbub Hasan, Adnan Mannan.

**Formal analysis:** Afroza Akter Tanni, Md. Mahbub Hasan, Adnan Mannan.

**Funding acquisition:** Adnan Mannan.

**Investigation:** Afroza Akter Tanni, Nahid Sultana, Wazir Ahmed, Adnan Mannan.

**Methodology:** Afroza Akter Tanni.

**Project administration:** Afroza Akter Tanni, Nahid Sultana, Adnan Mannan.

**Resources:** Md. Mahbub Hasan, Nahid Sultana, Wazir Ahmed.

**Software:** Md. Mahbub Hasan.

**Supervision:** Md. Mahbub Hasan, Nahid Sultana, Wazir Ahmed, Adnan Mannan.

**Validation:** Adnan Mannan.

**Visualization:** Afroza Akter Tanni, Wazir Ahmed.

**Writing – original draft:** Afroza Akter Tanni.

**Writing – review & editing:** Md. Mahbub Hasan, Adnan Mannan.

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
