## [Decision Letter · Decision Letter 0]

18 Jun 2021

PONE-D-21-08990

Prevalence of antibiotic resistance and associated genes among clinical isolates of Klebsiella pneumoniae in the southern region of Bangladesh

PLOS ONE

Dear Dr. Mannan,

Thank you for submitting your manuscript to PLOS ONE. After careful consideration, we feel that it has merit but does not fully meet PLOS ONE’s publication criteria as it currently stands. Therefore, we invite you to submit a revised version of the manuscript that addresses the points raised during the review process.

After careful consideration we had decided to provide you the opportunity to revise the manuscript as "major revisions". Please pay special attention to the comments provided by one of the reviewers as this reviewer felt it was a lot of room for improvement.

We look forward to see your revised version 

We look forward to receiving your revised manuscript.

Kind regards,

Monica Cartelle Gestal, PhD

Academic Editor

PLOS ONE

“This study was partially funded by Research and publication division, University of Chittagong. It was funded to AM.”

“Authors would like to thank the research assistants of Disease Biology and Moelcualr Epidemiology Research Group, Chattogram for their support during the study. This study was partially funded by Research and publication division, University of Chittagong.

“Authors would like to thank the research assistants of Disease Biology and Moelcualr Epidemiology Research Group, Chattogram for their support during the study. This study was partially funded by Research and publication division, University of Chittagong."

Additional Editor Comments (if provided):

Reviewers' comments:

Reviewer's Responses to Questions

**Comments to the Author**

1. Is the manuscript technically sound, and do the data support the conclusions?

Reviewer #1: Partly

Reviewer #2: Yes

Reviewer #3: Yes

2. Has the statistical analysis been performed appropriately and rigorously? 

Reviewer #1: N/A

Reviewer #2: Yes

Reviewer #3: N/A

3. Have the authors made all data underlying the findings in their manuscript fully available?

Reviewer #1: Yes

Reviewer #2: Yes

Reviewer #3: Yes

4. Is the manuscript presented in an intelligible fashion and written in standard English?

Reviewer #1: No

Reviewer #2: Yes

Reviewer #3: Yes

5. Review Comments to the Author

Reviewer #1: The manuscript by Tanni et al. investigated the prevalence of antibiotic resistance and among clinical isolates of K. pneumoniae in Bangladesh. Although I considered the results important under an epidemiological perspective, I regret to say that it presents many weaknesses.

- The are many mistyping through the text

- Resistance nomenclature should also be revised.

- Why did the authors specifically/only targeted the blaNDM-1 and blaSHV-11 genes? Although I recognize that blaNDM-1 is important in this global region, other clinically important ESBL and carbapenemase genes can support the phenotypic results.

- The authors used PCR amplification to identify the blaNDM-1 and blaSHV-11 genes, however more robust data (e.g., resistome, virulome, etc…) could be achieved by using the WGS approach.

- Conclusion sheds light on ESBL, but New Delhi Metallo-β-lactamase-1 (NDM-1) is undoubtedly more important.

Reviewer #2: It is suggested to do a review of the writing, some typographical errors were found. Review the discs used for the sensitivity study. A pie chart might give a better presentation in some cases. There is a duplication when uploading the graphics

Reviewer #3: Dear author,

Please, achieve next suggestions!

Methods:

2.3 Confirmation of K. pneumoniae (KPn) isolates

Please, specify the method used to identify the isolates

Results:

Table 1: Clinical characteristics of study population (n = 1000)

Please, could you divide the age group og 45, into 45-60 and >60. Thanks

Please, improve the figure, the quality and pixeles.

Thanks

6. PLOS authors have the option to publish the peer review history of their article (what does this mean?). If published, this will include your full peer review and any attached files.

Reviewer #1: No

Reviewer #2: No

Reviewer #3: No

---

## [Author Response · Author response to Decision Letter 0]

9 Jul 2021

Response to the reviewer

Reviewer #1:

The manuscript by Tanni et al. investigated the prevalence of antibiotic resistance and among clinical isolates of K. pneumoniae in Bangladesh. Although I considered the results important under an epidemiological perspective, I regret to say that it presents many weaknesses.

Response: The authors would like to thank the reviewer for pointing out the issues and helping us to improve its quality.

- The are many mistyping through the text

Response: we would like to express our apologies for typing errors. In the current format we have checked and corrected the typos. 

- Resistance nomenclature should also be revised.

Response: Resistance nomenclature has been changed both by definition and alphabetic order (Page 8 and Table 1). In table-1 a multidrug resistant column has been placed according to the standard definition by AP Magiorakos et al. 2021 and ESPAUR (https://assets.publishing.service.gov.uk/government/uploads/system/uploads/attachment_data/file/936199/ESPAUR_Report_2019-20.pdf). In the methodology, names of the antibiotics have been reorganized (Line-167, Page-8).

- Why did the authors specifically/only targeted the blaNDM-1 and blaSHV-11 genes? Although I recognize that blaNDM-1 is important in this global region, other clinically important ESBL and carbapenemase genes can support the phenotypic results.

Response: We would like to thank the reviewer for raising the issue. In this study we have found that the KPN isolates were resistant to more than one class of antibiotics available in the local market, regardless of the age group or type of samples. Then we wanted to find the molecular cause of the resistance in this area which is the second densely populated city in Bangladesh. Based on work that has been done on KPN in Bangladesh, we readily decided to assess the burden of blaNDM-1 and blaSHV-11. Importantly, the prevalence of the blaNDM-1 itself can determine the spread of deadly KPN that is evolving lately. However, for understanding the total scenario an in-depth molecular study is required and we are currently designing to do so.

- The authors used PCR amplification to identify the blaNDM-1 and blaSHV-11 genes, however more robust data (e.g., resistome, virulome, etc…) could be achieved by using the WGS approach.

Response: We agree that the WGS approach is more robust data to confirm the presence of a gene. However, PCR is also a state of the art technique in molecular biology and more convenient to conduct surveillance studies in a resource-limited country like Bangladesh. To check the validity of PCR data, in this study we have randomly checked 15 PCR amplicons by sequencing and found them accurate.

- Conclusion sheds light on ESBL, but New Delhi Metallo-β-lactamase-1 (NDM-1) is undoubtedly more important.

Response: We would like to thank the reviewer for this valuable suggestion. It has been added (Page 20) and conclusion has been modified as- 

In the K. pneumoniae infection landscape, our study was the first to run an extensive molecular analysis on ESBL producing K. pneumoniae isolates from patients seeking medical care in different hospitals in Chattogram, Bangladesh. Through this study, we were able to identify early occurrences of NDM-1 producing organisms as well as prevalence of blaNDM-1 plasmid. The prevalence of blaNDM-1 plasmid indicates a large-scale distribution of NDM-1 producing bacteria in the community. A substantial increase in resistant KPN strains in the environment, as well as in a healthcare setting, increases the possibility of nosocomial infections; and through horizontal gene transfer, it could also amplify the persistence of the whole antibiotic-resistance situation. Rigorous screening for NDM-1, SHV-11, UGE and other MBLs in ESBL-producing organisms, because it provides insight into a more genotypic aspect on KPN antibiotic resistance along with significant epidemiological factors will help to formulate and implement plans for controlling the use and abuse of antibiotics on a large scale as well as substitute treatment options in resistant cases.

References: 

Magiorakos, A. P., Srinivasan, A., Carey, R. B., Carmeli, Y., Falagas, M. E., Giske, C. G., ... & Monnet, D. L. (2012). Multidrug-resistant, extensively drug-resistant and pandrug-resistant bacteria: an international expert proposal for interim standard definitions for acquired resistance. Clinical microbiology and infection, 18(3), 268-281.

Reviewer #2:

It is suggested to do a review of the writing, some typographical errors were found. 

Response: We would like to thank the reviewer for this suggestion. We have rechecked the manuscript and corrected the errors. 

Review the discs used for the sensitivity study. A pie chart might give a better presentation in some cases. 

Response: We have deleted the information of antibiotics those have been suggested by the reviewers. Moreover, Another chart has been added where the information of sensitivity has been mentioned. A pie chart has been added (Figure 5) where the prevalence of antibiotic resistant genes in different samples has been shown.

There is a duplication when uploading the graphics

Response: Duplicate figures have been removed. 

Reviewer #3:

Dear author, Please, achieve next suggestions!

Methods:

2.3. Confirmation of K. pneumoniae (KPN) isolates. Please, specify the method used to identify the isolates.

Response: Initially, the bacterial colony and shape was checked in Mac-Conkey agar medium, Blood agar and Chocolate agar. Then, gram staining was done after culture for bacterial group differentiation. Thirdly, a number of conventional biochemical tests were performed [Indole, Methyl red, Voges Proskauer, Citrate, TSI (triple sugar Iron, Motility and Urease tests)] after subculture for Klebsiella pneumoniae isolates confirmation.This information has been added in the modified manuscript. 

First, we checked the bacterial colony and shape in MacConkey agar medium, blood agar, chocolate agar and L.B agar medium. Second, we did gram staining after culture for bacterial group differentiation. Thirdly, some conventional biochemical tests were performed (Indole, Methyl red, Voges Proskauer, Citrate, TSI (triple sugar Iron, Motility and Urease tests) for Klebsiella pneumoniae isolates confirmation. Finally, we selected 100 samples for molecular characterization of resistant determinants (blaNDM-1, blaSHV-11, blauge.). Finally, a phylogram was constructed with the sequence data in comparision with other KPN data retrieved from NCBI. 

Results:

Table 1: Clinical characteristics of study population (n = 1000)

Please, could you divide the age group og 45, into 45-60 and >60. Thanks

Response: Thank you for your valuable suggestion. The table has been revised and age group has been divided accordingly (Table 1).

Please, improve the figure, the quality and pixeles. Thanks

Response: Thank you for your valuable suggestion. The figure quality has been improved and the dpi of the modified figure is more than the previous version (300 dpi).

Reviewer’s recommendations

Line 84. When referring to the classes of antibiotics, ESBL is placed, which is a type of resistance and not a class of antibiotic. It is suggested to place beta-lactams as a class of antibiotics 

Response: changed ESBL with beta-lactams accordingly. 

Line 107. It is not necessary to capitalize Extended 

Response: corrected accordingly. 

Line 154. It is not necessary to capitalize Standard 

Response: amended accordingly. 

Lines 155 and 156. It is suggested to place the size of the Petri dishes used since 8 discs should not be placed in boxes of 100 x 10 (15) but a maximum of 6 according to the CLSI recommendations. 

Response: We have used 5 or 6 discs in a 90 mm plate. All the recommendations of CLSI (2021) were rechecked and followed. Two of the figures of the study have been attached below (Figure 1). 

Figure 1: Disc diffusion test for antibiotic sensitivity assessment.

Line 156. Why do you use Ampicillin disc for Klebsiella? Has natural resistance to this antibiotic 

Response: As this is a 600 bed hospital the discs were prepared for the need of all the patients as per requirements for other growths.

Intrinsic resistance to ampicillin is obvious in KPN. Sometimes, KPN carrying beta lactamase genes like blaSHV-11 show different sensitivity patterns to ampicillin (Kim J et al., 2016; Fu Y et al., 2007). Thus, to characterize the sensitivity pattern in different patients we used ampicillin. 

Line 157. Why do you place an azithromycin disk? This is used in S.Typhi or Shigella 

Response: Herein, we aimed to assess the resistance profile of KPN isolates which were isolated from various clinical and environmental sources against common antibiotics classes. Azithromycin is a bacterial protein synthesis inhibitor of the drug class, Erythromycin and also used for treating intestinal infections. Since, KPN is also an intestinal bacteria we used this disc for susceptibility testing. Moreover, there are previous studies which also mentioned the use of Azithromycin against KPN (Baker et. al. 2019). However, it has been removed from the list and figures of modified manuscript. 

The list of antibiotics used for the test does not include any beta-lactamase inhibitor. Why? 

Response: We used tazobactam/ piperacillin and amoxicillin/clavulanic acid as beta lactamase inhibitor. We did not mention these in description, because the percentage of resistance or sensitivity was not significant compared to other antibiotics. It has been included in raw data. However, it has also been added to the revised manuscript. 

A resistance analysis should be done according to the origin of the patient. Inpatients or outpatients 

Response: Resistance for inpatient and outpatient has been analyzed in the revised draft. A figure containing the data has been added in the modified draft (Figure 2, page-26). 

Line 214. What was the selection methodology for resistant Klebsiella? 

Response: We selected the resistant isolates based on their antimicrobial resistant profile. Multidrug resistant samples were selected for molecular analysis. Isolates showing resistance to the highest number of antibiotics were chosen for further analysis. Isolates that had antibiotic resistant genes were further sent for gene sequencing. It has been added in the modified manuscript (line- 172, page-8). 

Line 245, 246. The names of the bacteria must be written in italics 

Response: Amended accordingly. 

Line 252. Correct typing error 

Response: Amended accordingly. 

Line 263. Place a period after the word role 

Response: Amended accordingly.

Line 268. Correct typing error 

Response: Amended accordingly. 

Line 280. When speaking of a reservoir, it refers to a normal growth site. It is suggested to modify the word by site of infection 

Response: It has been modified accordingly.

Line 282. The year of the reference should be in parentheses 

Response: Amended accordingly.

Line 284 to 286. How did they show that they are treating the same Klebsiella clone as the patients infections? 

Response: In this study, Plasmid profiling showed a similarity between fomites and clinical isolates. Moreover, gene sequencing analyses showed that KPC-2 producing KPN strains were transferred from water to hospital and vice versa through mechanism. This study investigated the clonal relationship of the strains through phylogram and MLST. A number of studies also previously reported similar findings (Clarivet B, et al. 2017, Nascimento T. et al. 2017). However, it has not been mentioned as a conclusive statement in the revised manuscript. This was stated as a probable explanation (Line-342, Page- 17)-

In this study, we considered fomites for analyzing KPN isolates as well. We observed the presence of KPN isolates in sewerage water and bed trails from samples obtained from the neonatal ward. This indicates that not only clinical specimens but fomites also act as sites of infection of KPN [48, 49]. Presence of KPN in non-biological samples might be a probable explanation of the phenomenon of horizontal transfer of transposable elements into KPN isolates and rapid transmission of a plasmid to another Enterobacteriaceae.

Line 288. Write in italics Enterobacteriacea 

Response: Amended accordingly. 

Line 304. Remove the quotation mark after patients 

Response: Amended accordingly. 

Line 309. Put the year in parentheses 

Response: Amended accordingly. 

Line 315. Place the year in parentheses in the reference 

Response: Amended accordingly. 

Line 323. Enter the year of the reference 

Response: Amended accordingly. 

Line 327. Put the year in parentheses 

Response: Amended accordingly. 

Line 345. Correct typing error 

Response: Amended accordingly. 

Line 347. The study was carried out in two healthcare centers in Chattogram. The results could not be generalized to the southern Bangladesh region 

Response: Authors would like to thank the reviewer for this important suggestion. Title has been modified. New title reads “Prevalence and molecular characterization of antibiotic resistance and associated genes in Klebsiella pneumoniae isolates: A clinical observational study in different hospitals in Chattogram, Bangladesh”. 

Line 355. Correct typing error 

Response: amended accordingly. 

Figure 1A. Azithromycin should not be listed. AmoxiClav is not described in materials 

Response: Azithromycin has been used commonly in KPN infections and antibiotic sensitivity tests in Bangladesh (Jasmin et al., 2014). In our study, it was according to the common practice of the hospitals of Bangladesh and patients' needs. In addition to that, it was used in KPN infections in some trials according to literature studies (Getanda P et al., 2021). However, Azithromycin has been removed from the figure and the list in the modified draft. 

Amoxiclav has been mentioned in the methodology in the revised manuscript (page-8, Line-171). 

Figures 1-4 are repeated in the document 

Response: It has been deleted. Sorry for this inconvenience. 

References: 

Baker, K. R., Jana, B., Hansen, A. M., Vissing, K. J., Nielsen, H. M., Franzyk, H., & Guardabassi, L. (2019). Repurposing azithromycin and rifampicin against Gram-negative pathogens by combination with peptide potentiators. International journal of antimicrobial agents, 53(6), 868-872.

Clarivet B, Grau D, Jumas-Bilak E, Jean-Pierre H, Pantel A, Parer S, et al. Persisting transmission of carbapenemase-producing Klebsiella pneumoniae due to an environmental reservoir in a university hospital, France, 2012 to 2014. Eurosurveillance. 2016;21(17):30213.

Fu Y, Zhang F, Zhang W, Chen X, Zhao Y, Ma J, Bao L, Song W, Ohsugi T, Urano T, Liu S. Differential expression of bla(SHV) related to susceptibility to ampicillin in Klebsiella pneumoniae. Int J Antimicrob Agents. 2007 Mar;29(3):344-7. doi: 10.1016/j.ijantimicag.2006.10.015. Epub 2007 Feb 2. PMID: 17276039.

Getanda, P., Bojang, A., Camara, B., Jagne-Cox, I., Usuf, E., Howden, B. P., ... & Roca, A. (2021). Short-term increase in the carriage of azithromycin-resistant Escherichia coli and Klebsiella pneumoniae in mothers and their newborns following intra-partum azithromycin: a post hoc analysis of a double-blind randomized trial. JAC-Antimicrobial Resistance, 3(1), dlaa128.

Islam MA, Talukdar PK, Hoque A, Huq M, Nabi A, Ahmed D, Talukder KA, Pietroni MA, Hays JP, Cravioto A, Endtz HP. Emergence of multidrug-resistant NDM-1-producing Gram-negative bacteria in Bangladesh. Eur J Clin Microbiol Infect Dis. 2012 Oct;31(10):2593-600. doi: 10.1007/s10096-012-1601-2. Epub 2012 Mar 17. PMID: 22422273.

Kim, J., Jo, A., Chukeatirote, E., & Ahn, J. (2016). Assessment of antibiotic resistance in Klebsiella pneumoniae exposed to sequential in vitro antibiotic treatments. Annals of clinical microbiology and antimicrobials, 15(1), 60. https://doi.org/10.1186/s12941-016-0173-x

Jesmin Akter, A.M. Masudul Azad Chowdhury and Mohammad Al Forkan , 2014. Study on Prevalence and Antibiotic Resistance Pattern of Klebsiella Isolated from Clinical Samples in South East Region of Bangladesh. American Journal of Drug Discovery and Development, 4: 73-79.

Magiorakos, A. P., Srinivasan, A., Carey, R. B., Carmeli, Y., Falagas, M. E., Giske, C. G., ... & Monnet, D. L. (2012). Multidrug-resistant, extensively drug-resistant and pandrug-resistant bacteria: an international expert proposal for interim standard definitions for acquired resistance. Clinical microbiology and infection, 18(3), 268-281.

Nascimento T, Cantamessa R, Melo L, Fernandes MR, Fraga E, Dropa M, et al. International high-risk clones of Klebsiella pneumoniae KPC-2/CC258 and Escherichia coli CTX-M-15/CC10 in urban lake waters. Science of The Total Environment. 2017;598:910-5.

Pauline Getanda, Abdoulie Bojang, Bully Camara, Isatou Jagne-Cox, Effua Usuf, Benjamin P Howden, Umberto D’Alessandro, Christian Bottomley, Anna Roca, Short-term increase in the carriage of azithromycin-resistant Escherichia coli and Klebsiella pneumoniae in mothers and their newborns following intra-partum azithromycin: a post hoc analysis of a double-blind randomized trial, JAC-Antimicrobial Resistance, Volume 3, Issue 1, March 2021, dlaa128, https://doi.org/10.1093/jacamr/dlaa128

---

## [Decision Letter · Decision Letter 1]

4 Aug 2021

PONE-D-21-08990R1

Prevalence and molecular characterization of antibiotic resistance and associated genes in Klebsiella pneumoniae isolates: A clinical observational study in different hospitals in Chattogram, Bangladesh

PLOS ONE

Dear Dr. Mannan,

Thank you for submitting your manuscript to PLOS ONE. After careful consideration, we feel that it has merit but does not fully meet PLOS ONE’s publication criteria as it currently stands. Therefore, we invite you to submit a revised version of the manuscript that addresses the points raised during the review process.

Please, address the comments provider by the reviewer and resubmit as early as your convenience

We look forward to receiving your revised manuscript.

Kind regards,

Monica Cartelle Gestal, PhD

Academic Editor

PLOS ONE

Journal Requirements:

Additional Editor Comments (if provided):

Reviewers' comments:

Reviewer's Responses to Questions

**Comments to the Author**

1. If the authors have adequately addressed your comments raised in a previous round of review and you feel that this manuscript is now acceptable for publication, you may indicate that here to bypass the “Comments to the Author” section, enter your conflict of interest statement in the “Confidential to Editor” section, and submit your "Accept" recommendation.

Reviewer #1: All comments have been addressed

Reviewer #2: All comments have been addressed

2. Is the manuscript technically sound, and do the data support the conclusions?

Reviewer #1: Yes

Reviewer #2: Yes

3. Has the statistical analysis been performed appropriately and rigorously? 

Reviewer #1: N/A

Reviewer #2: Yes

4. Have the authors made all data underlying the findings in their manuscript fully available?

Reviewer #1: Yes

Reviewer #2: Yes

5. Is the manuscript presented in an intelligible fashion and written in standard English?

Reviewer #1: Yes

Reviewer #2: Yes

6. Review Comments to the Author

Reviewer #1: (No Response)

Reviewer #2: The article sent for review has been improved in relation to the first version, however, some minor recommendations are made to improve it.

7. PLOS authors have the option to publish the peer review history of their article (what does this mean?). If published, this will include your full peer review and any attached files.

Reviewer #1: No

Reviewer #2: No

---

## [Author Response · Author response to Decision Letter 1]

6 Aug 2021

We thank the honorable reviewer for valuable feedback. Please find our response below-

Line 41. Klebsiella in italics

Response: We would like to thank the reviewer for this suggestion. We have rechecked the manuscript and corrected the error. 

Line 82. Place the year in parentheses

Response: We would like to thank the reviewer for this suggestion. We have rechecked the manuscript and corrected the error. 

Line 87. Enterobacteriaceae in italics

Response: We would like to thank the reviewer for this suggestion. We have rechecked the manuscript and corrected the error. 

Lines 147 to 150. They are proper names of the units and the first letter of each word should 

be capitalized (unify, some are capitalized while others are not)

Response: We would like to thank the reviewer for this suggestion. We have rechecked the manuscript and edited accordingly. 

Line 170. Klebsiella pneumoniae in italics

Response: We would like to thank the reviewer for this suggestion. We have rechecked the manuscript and corrected the error. 

Line 186 to 191. When describing the primers used, it is not necessary to place F or R before the sequence, taking into account that they already indicate whether it is Forward or Reverse. It is recommended to place the name of the amplified gene in the case of the third pair of primers. If the 5'-3' orientation is specified at the beginning of all primers, the extremities of each sequence could be eliminated. All this in order to improve the presentation and facilitate reading.

Response: We would like to thank the reviewer for this suggestion. Changes have been made accordingly. 

Line 192 the reference has an asterisk. Can't find where this asterisk points to

Response: We would like to thank the reviewer for this suggestion. We have rechecked the manuscript and corrected the error. 

Line 265 Correct writing ceftriaxoMe

Response: We would like to thank the reviewer for this suggestion. We have rechecked the manuscript and corrected the error. 

Line 277 separate Figure 3; Table2. To avoid confusion

Response: We would like to thank the reviewer for this suggestion. We have rechecked the manuscript and edited accordingly. 

Figure 4. It would be interesting for the authors to mention the clonality of these analyzes. Try to hypothesize against this. The analysis in sewerage water and bed trails from samples obtained from the neonatal Ward is mentioned. Did you make comparative analyzes between the KPN of samples with those of fomites? Was cross contamination found ?. It is recommended to discuss and conclude regarding the clonality found, this data is of interest

Response: We agree with the reviewer that clonality should be monitored. However, the fomites were assessed only to check the presence of KPN. It can give an idea if there is a chance of transmission / cross contamination from fomite to the patients. But this needs more samples from fomites. And an in depth molecular study on fomite and source of contamination is needed. We are working on the molecular identification of fomite and reservoir of the pathogen in the next phase of this study and that is a separate project. Considering the need of in depth analysis, we have excluded this part from the result and discussion of the modified manuscript. We believe this will help the readers to keep focus on the prevalence of antibiotic resistance in Klebsiella isolates from the clinic samples. 

Line 365 to 370. Fomite analysis is not mentioned in materials and methods.

Response: Information regarding fomite has been excluded from the modified manuscript and the reason for this is explained in the above response. 

Line 404. Place the year in parentheses

Response: We would like to thank the reviewer for this suggestion. We have rechecked the manuscript and corrected the error. 

Line 429. Genus and bacterial species in italic

Response: We would like to thank the reviewer for this suggestion. We have rechecked the manuscript and corrected the error.

---

## [Decision Letter · Decision Letter 2]

1 Sep 2021

Prevalence and molecular characterization of antibiotic resistance and associated genes in Klebsiella pneumoniae isolates: A clinical observational study in different hospitals in Chattogram, Bangladesh

PONE-D-21-08990R2

Dear Dr. Mannan,

We’re pleased to inform you that your manuscript has been judged scientifically suitable for publication and will be formally accepted for publication once it meets all outstanding technical requirements.

Kind regards,

Monica Cartelle Gestal, PhD

Academic Editor

PLOS ONE

Additional Editor Comments (optional):

Reviewers' comments:

Reviewer's Responses to Questions

**Comments to the Author**

1. If the authors have adequately addressed your comments raised in a previous round of review and you feel that this manuscript is now acceptable for publication, you may indicate that here to bypass the “Comments to the Author” section, enter your conflict of interest statement in the “Confidential to Editor” section, and submit your "Accept" recommendation.

Reviewer #1: All comments have been addressed

Reviewer #2: All comments have been addressed

2. Is the manuscript technically sound, and do the data support the conclusions?

Reviewer #1: Yes

Reviewer #2: Yes

3. Has the statistical analysis been performed appropriately and rigorously? 

Reviewer #1: Yes

Reviewer #2: Yes

4. Have the authors made all data underlying the findings in their manuscript fully available?

Reviewer #1: Yes

Reviewer #2: Yes

5. Is the manuscript presented in an intelligible fashion and written in standard English?

Reviewer #1: Yes

Reviewer #2: Yes

6. Review Comments to the Author

Reviewer #1: The authors clarified all questions that I pointed out and addressed all my concerns. I have no additional comments.

Reviewer #2: After reviewing the requested changes, the article is considered ready for publication. Congratulations to the authors for their effort and dedication.

7. PLOS authors have the option to publish the peer review history of their article (what does this mean?). If published, this will include your full peer review and any attached files.

Reviewer #1: No

Reviewer #2: No

---

## [Editor Report · Acceptance letter]

3 Sep 2021

PONE-D-21-08990R2 

Prevalence and molecular characterization of antibiotic resistance and associated genes in *Klebsiella pneumoniae* isolates: A clinical observational study in different hospitals in Chattogram, Bangladesh 

Dear Dr. Mannan:

I'm pleased to inform you that your manuscript has been deemed suitable for publication in PLOS ONE. Congratulations! Your manuscript is now with our production department. 

Kind regards, 

on behalf of

Dr. Monica Cartelle Gestal 

Academic Editor

PLOS ONE